# Spiking Convolutional Neural Networks for Text Classification

**Changze Lv, Jianhan Xu, and Xiaoqing Zheng**[*]
School of Computer Science, Fudan University, Shanghai 200433, China
Shanghai Key Laboratory of Intelligent Information Processing
{czlv18,jianhanxu20,zhengxq}@fudan.edu.cn

## Abstract

Spiking neural networks (SNNs) offer a promising pathway to implement deep neural networks (DNNs) in a more energy-efficient manner since their neurons are sparsely activated and inferences are event-driven. However, there have been very few works that have demonstrated the efficacy of SNNs in language tasks partially because it is non-trivial to represent words in the forms of spikes and to deal with variable-length texts by SNNs. This work presents a "conversion + fine-tuning" two-step method for training SNNs for text classification and proposes a simple but effective way to encode pre-trained word embeddings as spike trains. We show empirically that after fine-tuning with surrogate gradients, the converted SNNs achieve comparable results to their DNN counterparts with much less energy consumption across multiple datasets for both English and Chinese. We also show that such SNNs are more robust to adversarial attacks than DNNs.

## 1 Introduction

Inspired by the biological neuro-synaptic framework, modern deep neural networks are successfully used in various applications (Krizhevsky et al., 2012; Graves & Jaitly, 2014; Mikolov et al., 2013b). However, the amount of computational power and energy required to run state-of-the-art deep neural models is considerable and continues to increase in the past decade. For example, a neural language model of GPT-3 (Brown et al., 2020) consumes roughly $190,000$ kWh to train (Dhar, 2020; Anthony et al., 2020), while the human brain performs perception, recognition, reasoning, control, and movement simultaneously with a power budget of just $20$ W (Cox & Dean, 2014). Like biological neurons, spiking neural networks (SNNs) use discrete spikes to compute and transmit information, which are more biologically plausible and also energy-efficient than deep learning models. Spike-based computing fuelled with neuromorphic hardware provides a promising way to realize artificial intelligence while greatly reducing energy consumption.

Although many studies have shown that SNNs can produce competitive results in vision (mostly classification) tasks (Cao et al., 2015; Diehl et al., 2015; Rueckauer et al., 2017; Shrestha & Orchard, 2018; Sengupta et al., 2019), there are very few works that have demonstrated their effectiveness in natural language processing (NLP) tasks (Diehl et al., 2016; Rao et al., 2022). SNNs offer a promising opportunity for processing sequential data. Rao et al. (2022) showed that long-short term memory (LSTM) units can be implemented by spike-based neuromorphic hardware with the spike frequency adaptation mechanism. They tested the performance of such spike-based networks (called RelNet) on a question-answering dataset (Weston et al., 2015), and observed that RelNet could solve 16 out of the 17 toy tasks. A task is considered to be solved if the network has an error rate at most $5\%$ on unseen instances of the task. In their design, each word is encoded as a one-hot vector, and a sentence is also fed into the network in the form of one-hot coded spikes. Such a one-hot encoding schema limits the size of the vocabulary that could be used (otherwise, very high-dimensional vectors are required to represent words used in a language). Besides, it is impossible for spike-based networks to leverage the word embeddings learned from a large amount of text data. Diehl et al. (2016) used pre-trained word embeddings in their TrueNorth implementation of a recurrent neural network and achieved $74\%$ accuracy in a question classification task. However, an external projection layer is required to project word embeddings to the vectors with positive values that can be further converted

---

[*]Corresponding author

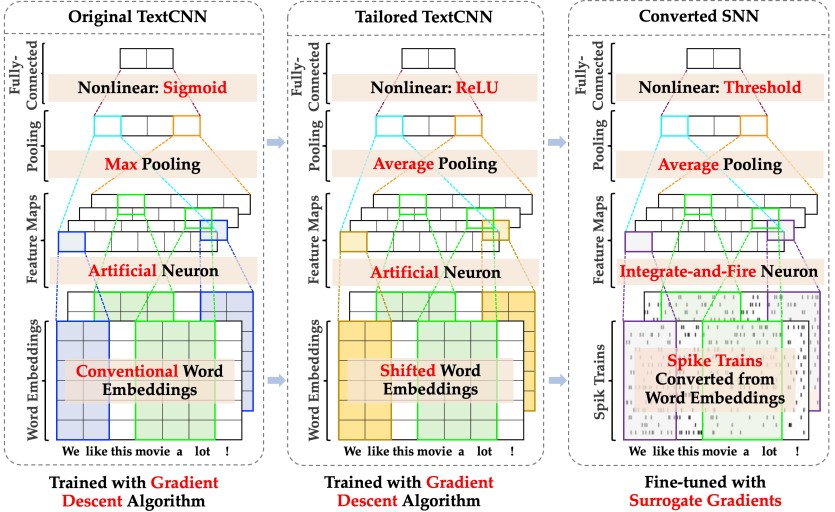

Figure 1: An illustration of a two-step method (conversion + fine-tuning) for training spiking neural networks for text classification: initialize an SNN with the weights of a tailored network trained with the gradient descent, and perform backpropagation with surrogate gradients on the converted SNN. The tailored network is obtained by replacing the max-pooling operation with average-pooling, the Sigmoid activation function with ReLU, and the word embeddings with their positive equivalents.

into spike trains. Such a projection layer cannot be easily implemented by spike-based networks, and in fact, they used a hybrid architecture that combines artificial and spiking neural networks.

In this study, we propose a two-step recipe of "conversion + fine-tuning" to train spiking neural networks for NLP. A normally-trained neural network is first converted to a spiking neural network by simply duplicating its architecture and weights, and then the converted SNN is fine-tuned afterward. Before the conversion, a proper tailored network needs to be built and trained first. Taking a convolutional neural network for sentence classification, called TextCNN (Kim, 2014), as an example (see Figure 1), the original TextCNN is first modified to a tailored CNN by replacing the max-pooling operation with average-pooling, the Sigmoid activation function with ReLU, and the word embeddings with positive-valued vectors (shifted). After the tailored network is trained on a dataset with the gradient descent algorithm, it is converted to a spiking neural network that is further fine-tuned with the surrogate gradient method (Zenke & Vogels, 2021) on the same datast. The SNNs trained with the proposed two-step training strategy yield comparable results to their DNN counterparts with much less energy consumption. The contribution of this study can be summarized as follows:

- We present a two-step method for training SNNs for language tasks, which combines the conversion-based approach (also known as shallow training) and the backpropagation using surrogate gradients at the fine-tuning phase.
- We propose a method to convert word embeddings to spike trains, which makes it possible for SNNs to leverage the word embeddings pre-trained from a large amount of text data. The ablation study shows that using pre-trained word embeddings can significantly improve the performance of SNNs.
- This study is among the first to demonstrate that well-trained spiking neural networks can achieve comparable results to their DNN counterparts on 6 text classification datasets and for both English and Chinese languages. We also show that SNNs perform more robustly against adversarial attacks than traditional DNNs.

## 2 RELATED WORK

SNNs offer a promising computing paradigm due to their ability to capture the temporal dynamics of biological neurons. Several methods have been proposed for training SNNs, and they can be roughly divided into two categories: conversion-based and spike-based approaches. The conversion-based approaches are to train a non-spiking network first and convert it into an SNN that produces the same input-output mapping for a given task as that of the original network. In the spike-based approaches, SNNs are trained using spike-timing information in an unsupervised or supervised manner.

The advantage of conversion-based approaches is that the non-differentiability of discrete spikes can be circumvented and the burden of training in the temporal domain is partially removed. Cao et al. (2015) proposed an approach for converting a deep CNN into an SNN by interpreting the activations as firing rates. To minimize performance loss in the conversion process, Diehl et al. (2015) presented a new weight normalization method to regulate firing rates, which boosts the performance of SNNs without additional training time. Sengupta et al. (2019) pushed spiking neural networks going deeper by exploring residual architectures and introducing a layer-by-layer weight normalization method. However, the conversion is just an approximation, leading to a decline in the accuracy of converted SNNs. Another drawback of such approaches is that converting high precision activations into spikes requires a long sequence of time steps in simulation which increases latency at the inference. Rathi et al. (2020) proposed a hybrid approach to partially address the issue of long-time sequences, which is most related to this study. Initialized with the weights from a trained neural network, the converted SNN is trained using backpropagation. Although this helps to reduce the number of required time steps, it appears to degrade accuracy in image classification tasks. In contrast, we experimentally show that the fine-tuning with surrogate gradients can further improve the accuracy across multiple text classification datasets and the obtained SNNs are capable of integrating the temporal dynamics of spikes properly derived from the pre-trained word embeddings via backpropagation through time.

Inspired by neuroscience, unsupervised SNN training with local STDP-based rules has drawn great attention (Masquelier et al., 2009). Diehl & Cook (2015) demonstrated that an SNN trained in a completely unsupervised way yields comparable accuracy to deep learning on the MNIST dataset. However, unsupervised trained SNNs generally perform worse than their supervised counterparts. Early works in supervised approaches are the tempotron (Gütig & Sompolinsky, 2006) and ReSuMe (Ponulak & Kasiński, 2010). Most works in this line rely on the gradients estimated by a differentiable approximate function so that gradient descent can be applied with backpropagation using spike times (Bohte et al., 2002; Booij & tat Nguyen, 2005) or backpropagation using spikes (i.e., backpropagation through time) (Shrestha & Orchard, 2018; Hunsberger & Eliasmith, 2015; Bellec et al., 2018; Huh & Sejnowski, 2018). To date, supervised learning has been unable to surpass the conversion-based approaches although it turns out to be more computationally efficient. SNNs have provided competitive results, but mostly in vision-related tasks. In this study, we provide proof of experiments that spiking CNNs can yield competitive accuracies over multiple language datasets by combining the advantages of conversion-based approaches and backpropagation using spikes.

## 3 METHOD

We describe our method for training SNNs for text classification in the following. We first present the approach to building tailored neural networks and the conversion process by taking TextCNN as an example (Kim, 2014), and then depict the way to fine-tune the converted SNNs with surrogate gradients, where the pre-trained word embeddings were transformed into spike trains produced by a Poisson event-generation function. The entire training procedure is summarized in Algorithm 1.

### 3.1 CONVERSION-BASED APPROACH

The idea behind the conversion-based approaches is simple—interpreting the activations as firing rates and mapping the magnitude of values output by each unit of a DNN to the frequency of spikes generated by the corresponding neuron of the converted SNN. To enable such conversions, a DNN architecture should be tailored to fit the requirements of SNN by removing some operations (listed in Subsection 3.1.1) that cannot be realized by spike-based computation (Cao et al., 2015). The tailored DNN is trained in the same way as one would with conventional DNN, and the learned weights are then applied to the SNN converted from the tailored DNN. Some weight normalization methods are often applied to regulate firing rates after the conversion (Diehl et al., 2015; Rueckauer et al., 2017; Sengupta et al., 2019). From our experimentation on multiple language datasets, we found that when the conversion is followed by the fine-tuning step, the effectiveness of weight normalization is negligible for text classification tasks (see Subsection 4.3).

### 3.1.1 TAILORED NEURAL NETWORK

Although the proposed method can be applied to all the deep neural architectures that can be converted to SNNs, we take the convolutional neural networks for sentence classification (TextCNN) (Kim, 2014) as an example architecture for clarity (this architecture is also used in the experiments). As shown in Figure 1, the TextCNN applies a convolution layer with multiple filter widths and

feature maps over word embeddings learnt from an unsupervised neural language model, and then summarizes the outputs of the convolution layer over time by a max-pooling (followed by a fully-connected layer) to produce a sentence representation. The TextCNN is trained over the summarized representations by minimizing a given loss function.

The operations that will produce negative values or those not supported by SNNs should be avoided or replaced with other alternatives in tailored neural networks because negative values are quite hard to be precisely represented in SNNs. There is also no simple and good way to implement the max-pooling operation in SNNs, which requires two additional network layers with lateral inhibition and causes a loss in accuracy due to the additional complexity. The biases of neurons also cannot easily be implemented in SNNs since their value could be positive or negative. Like Cao et al. (2015), we create tailored neural networks by making the following changes to their original DNNs:

- Word embeddings are converted into the vectors of the same dimension with positive values by normalization and shifting (discuss later in Subsection 3.1.2).
- All the non-linear activation functions are replaced with ReLU (rectified linear unit) activation function (i.e., $\text{ReLU}(x) = \max(x, 0)$).
- The biases are removed from all the convolutional and fully-connected layers.
- The average-pooling is used instead of the max-pooling, which can be easily implemented in spike-based computation.

### 3.1.2 PRE-TRAINED WORD EMBEDDINGS

Pre-trained word embeddings have been successfully used in a wide range of NLP tasks, and they should be useful for SNNs to generalize from a training set with a limited size to possible unseen texts too. The values of word embeddings are not all positive, and therefore an appropriate method is required to convert those word embeddings into the vectors with positive values so that the inputs to the first layer of an SNN are all non-negative. We tried several methods to fulfill this purpose, and found the following one is simple, but effective in preserving the semantic regularities in language captured in the original word embeddings when they are converted and transformed to spike trains. We first calculate the mean value $\mu$ and the standard deviation $\sigma$ of the values of pre-trained word embeddings, clip all the values within $[\mu + 3\sigma, \mu - 3\sigma]$, perform the normalization by subtracting $\mu$ and then dividing by $6 \times \sigma$, and shift all the components of vectors within the range of $[0, 1]$.

For English, we used 300-dimentional GloVe embeddings (Pennington et al., 2014) trained on a text dataset merged from Wikipedia 2014 and Gigaword 5. For Chinese, the Word2Vec toolkit (Mikolov et al., 2013a) was used to learn word embeddings of 300 dimensions from a text corpus composed of Baidu Encyclopedia, Wikipedia-zh, People's Daily News, Sogou News, Financial News, ZhihuQA, Weibo and Complete Library in Four Sections (Li et al., 2018b). In the supervised training stage of tailored neural networks, the converted (positive-valued) word embeddings are free to modify and their values will be clipped to $[0, 1]$ if they were modified to values greater than 1 or less than 0. SNNs only take spikes as input, and thus a Poisson spike train will be generated for each component of a word embedding with a firing rate proportional to its scale.

### 3.1.3 TRAINING AND CONVERSION

We use the cross-entropy loss to train the tailored neural networks as usual. As illustrated in Figure 1, the conversion of a tailored network to an SNN is straightforward. All the processing blocks of the converted SNN are inherited from the tailored network except a spike generator that is added to the SNN, which is used to generate Poisson spike trains derived from the learned word embeddings. Each neuron of the tailored network will be replaced with a leaky integrate-and-fire neuron (discuss later in subsection 3.2.1), and the weights for convolutional and fully-connected layers in the tailored network become the synaptic strengths in the converted SNN. The ReLU activation functions are no longer needed since their functionality is implicitly provided by the neuron's membrane threshold.

### 3.2 FINE-TUNING WITH SURROGATE GRADIENTS

Once an SNN is converted from a tailored network (trained), we can fine-tune the converted SNN by the generalized backpropagation algorithm with surrogate gradients on the same dataset that was used to train the tailored network. The converted weights can be viewed as a good initialization, which contributes to solving the problem of temporal and spatial credit assignment for the SNN. In

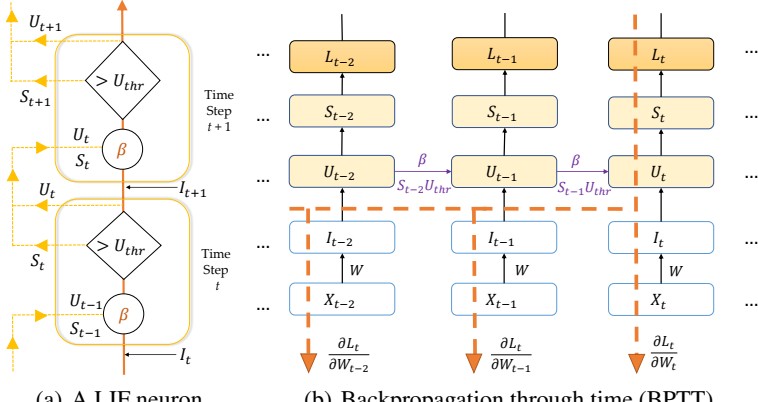

(a) A LIF neuron        (b) Backpropagation through time (BPTT)

Figure 2: Computational steps in training SNNs by the generalized backpropagation with surrogate gradients. (a) A recurrent representation of a leaky integrate-and-fire (LIF) neuron. (b) An unrolled computational graph of the LIF neuron where time flows from left to right.

the fine-tuning stage, the word embeddings present in the form of spike trains are fixed because such spike trains are generated randomly and there is no one-to-one mapping between a word embedding and its spike train. However, such temporal codes with timing disturbances make SNNs more robust to adversarial attacks (see the experimental results in subsection 4.4) because the trained SNNs are more tolerant to noise appearing in input spike trains randomly generated.

### 3.2.1 INTEGRATE-AND-FIRE NEURON

There have been many spiking neuron models that could be used to build SNNs (Izhikevich, 2004), we chose to use a widely-used, first-order leaky integrate-and-fire (LIF) neuron as the building block. Like the artificial neuron model, LIF neurons operate on a weighted sum of inputs, which contributes to the membrane potential $U_t$ of the neuron at time step $t$. If the neuron is sufficiently excited by the weighted sum and its membrane potential reaches a threshold $U_{\text{thr}}$, a spike $S_t$ will be generated:

$$S_t = \begin{cases} 1, & \text{if } U_t \geq U_{\text{thr}}; \\ 0, & \text{if } U_t < U_{\text{thr}}. \end{cases} \tag{1}$$

The dynamics of the neuron's membrane potential can be modelled as an resistor–capacitor circuit, an approximate solution to the differential equation of this circuit can be represented as follows:

$$U_t = I_t + \beta U_{t-1} - S_{t-1} U_{\text{thr}}$$
$$I_t = W X_t \tag{2}$$

where $X_t$ are inputs to the LIF neuron at time step $t$, $W$ is a set of learnable weights used to integrate different inputs, $I_t$ is the weighted sum of inputs, $\beta$ is the decay rate of membrane potential, and $U_{t-1}$ is the membrane potential at the previous time $t-1$. The last term of $S_{t-1} U_{\text{thr}}$ is introduced to account for spiking and membrane potential reset. A LIF neuron model is illustrated in Figure 2 (a).

### 3.2.2 SURROGATE GRADIENT

Backpropagation through time (BPTT) is one of the most popular approaches for training SNNs (Shrestha & Orchard, 2018; Huh & Sejnowski, 2018; Cramer et al., 2022). This approach applies the generalized backpropagation algorithm to the unrolled computational graph. The gradients flow from the final output of a network to its input layer (indicated by the orange dashed arrows in Figure 2 (b)). In this way, computing the gradients through an SNN is most similar to that of a recurrent neural network. To deal with the spike non-differentiability problem, the Heaviside step function, shown as Equation (1), is applied to determine whether a neuron emits a spike during the forward pass while this function is replaced with a differentiable one during the backward pass. The derivative of the differentiable function is used as a surrogate, and this approach is known as the surrogate gradient. We chose to use Fast-Sigmoid (Zheng & Mazumder, 2018) as the surrogate function $\hat{S}_t$, where $k$ is set to 25 by default:

$$\hat{S}_t \approx \frac{U_t}{1 + k|U_t|} \tag{3}$$

---

**Algorithm 1** The global algorithm of "conversion + fine-tuning" for training spiking neural networks.

---

**Input:** $E$: A set of pre-trained word embeddings;

$\quad\quad$ $D$: A training set consisting of $N$ instances $\{(x_i, y_i)\}_{i=1}^{N}$;

$\quad\quad$ $R$: A traditional neural network (architecture only);

$\quad\quad$ $M$: A spiking neural network with a set of learnable weights $W$;

$\quad\quad$ $\beta$: A decay rate of membrane potential;

$\quad\quad$ $U_{\text{thr}}$: A membrane threshold;

$\quad\quad$ $\eta$: A learning rate $\eta$ used at the fine-tuning phase.

**I. Conversion Step**:

$\quad$ Transform pre-trained word embeddings $E$ to the vectors with positive values (Subsection 3.1.2);

$\quad$ Build a tailored neural network from a traditional neural network $R$ (Subsection 3.1.1);

$\quad$ Train the tailored neural network on the training set $D$ with the gradient descent algorithm;

$\quad$ Convert the tailored neural network (trained) to the corresponding spiking neural network $M$.

**II. Fine-Tuning Step**:

$\quad$ **for** *each mini-batch $B$ in $D$* **do**

$\quad\quad$ Generate a Poisson spike train for each component of all the word embeddings appeared in $B$;

$\quad\quad$ Perform a forward pass and record the spikes as well as the membrane potentials at every time step;

$\quad\quad$ Calculate the derivative of the loss with respect to the weights (see Equation (4));

$\quad\quad$ Update the weights of the spiking neural network $M$ by $W = W - \eta \frac{\partial L}{\partial W}$ (see Appendix A.1).

**end**

**Return** The fine-tuned spiking neural network $M$.

---

### 3.2.3 Loss Function

Since the surrogate function, like Equation (3), is applied when working backward, we can add a softmax layer to the end of SNNs to predict the category labels at each time step $t$ for an instance $i$, denoted by $\hat{y}_t^i$. The SNNs are fine-tuned by minimizing the cross-entropy spike rate error using the generalized gradient descent. This is equivalent to minimizing the KL-divergence between the prediction distribution $\hat{y}_t^i$ and the target distribution $y^i$ at each time step $t$ for an instance $i$. We use the 1-of-$K$ coding scheme to represent the target $y^i$. The loss function for $N$ training instances is:

$$L = -\frac{1}{N}\sum_{i=1}^{N}\left(\frac{1}{T}\sum_{t=1}^{T}\left(y^i \times \log(\hat{y}_t^i)\right)\right) \tag{4}$$

where $T$ is the number of time steps used for training SNNs. Finding the derivative of this loss with respect to the weights allows the use of gradient decent to train SNNs. We list an efficient way to calculate the derivatives in Appendix A.1.

## 4 Experiments

We conducted four sets of experiments. The first is to evaluate the accuracy of the SNNs trained with the proposed method on 6 different text classification benchmarks for both English and Chinese by comparing to their DNN counterparts. The goal of the second experiment is to see how robust the SNNs would be to defend against existing sophisticated adversarial attacks. The third one is to show that the conversion and fine-tuning sets are essential for training SNNs by ablation study. The last experiment is to see how the performance of SNNs is impacted by the value of decay rate, the number of neurons used to predict for each category, and the value of membrane threshold.

### 4.1 Dataset

We used the following 6 text classification datasets to evaluate the SNNs trained with the proposed method, four of which are English datasets and the other two are Chinese benchmarks: MR (Pang & Lee, 2005), SST-5 (Socher et al., 2013), SST-2 (the binary version of SST-5), Subj, ChnSenti, and Waimai. These datasets vary in the size of examples and the length of texts. If there is no standard training-test split, we randomly select $10\%$ examples from the entire dataset as the test set. We describe the datasets used for evaluation in Appendix A.2.

Table 1: Classification accuracy achieved by different models on 6 datasets. The model obtained by applying the model-based normalization on the converted SNN is denoted as "Conv SNN + MN" and that by applying the data-based normalization as "Conv SNN + DN". The SNNs trained with the "conversion + fine-tuning" is denoted as "Conv SNN + FT".

| Method | English Dataset | | | | Chinese Dataset | |
|---|---|---|---|---|---|---|
| | MR | SST-2 | Subj | SST-5 | ChnSenti | Waimai |
| Original TextCNN | $77.41\pm0.22$ | $83.25\pm0.16$ | $94.00\pm0.22$ | $45.48\pm0.16$ | $86.74\pm0.15$ | $88.49\pm0.16$ |
| Tailored TextCNN | $76.94\pm0.25$ | $83.03\pm0.21$ | $91.50\pm0.12$ | $43.48\pm0.13$ | $85.79\pm0.15$ | $88.21\pm0.15$ |
| Directly-trained SNN | $51.55\pm1.31$ | $75.73\pm0.91$ | $53.30\pm1.80$ | $23.08\pm0.56$ | $63.18\pm0.42$ | $66.42\pm0.39$ |
| Conv SNN | $74.13\pm0.97$ | $80.07\pm0.78$ | $90.40\pm0.39$ | $41.40\pm0.73$ | $84.16\pm0.62$ | $86.43\pm0.43$ |
| Conv SNN + MN | $74.70\pm0.52$ | $79.90\pm0.61$ | $89.40\pm0.57$ | $40.59\pm1.13$ | $84.89\pm0.32$ | $85.21\pm0.46$ |
| Conv SNN + DN | $74.19\pm0.78$ | $80.67\pm0.95$ | $90.30\pm0.86$ | $40.63\pm1.78$ | $83.73\pm0.35$ | $86.33\pm0.35$ |
| Conv SNN + FT | $\mathbf{75.45}\pm0.51$ | $\mathbf{80.91}\pm0.34$ | $\mathbf{90.60}\pm0.32$ | $\mathbf{41.63}\pm0.44$ | $\mathbf{85.02}\pm0.22$ | $\mathbf{86.66}\pm0.17$ |

## 4.2 IMPLEMENTATION DETAILS

We used TextCNN (Kim, 2014) as the neural network architecture from which the tailored network is built, and filter widths of 3, 4, and 5 with 100 feature maps each. When training the tailored networks, we set the dropout rate to 0.5, the batch size to 32, and the learning rate to $1e - 4$.

SnnTorch framework provided by (Eshraghian et al., 2021) was used to train SNNs, which extends the capabilities of PyTorch (Paszke et al., 2019) and can perform gradient-based learning with SNNs. We set the number of time steps to 50, the membrane threshold $U_{\text{thr}}$ to 1, the decay rate $\beta$ to 1, the batch size to 50, and the learning rate to $5e - 5$ at the fine-tuning stage of SNNs.

Eshraghian et al. (2021) showed that if we collect the results produced by multiple neurons and count their spikes, it is possible to accurately measure a firing rate from a population of neurons in a very short time window. Therefore, we also used such a commonly-used ensemble method in the tailored networks and the converted SNNs. Specifically, instead of assigning one neuron to each category for prediction, we use $h$ neurons for each category and the prediction results on $h$ spiking neurons are ensembled to get a final output. Unless otherwise specified, we set $h$ to 10 in all the experiments.

## 4.3 RESULTS

We reported in Table 1 the classification accuracy achieved by the SNNs trained with the "conversion + fine-tuning" method on 4 English and 2 Chinese datasets, compared to several baselines, including the converted SNNs without the fine-tuning, the converted SNNs with two weight normalization methods, and the SNNs directly-trained with surrogate gradients without using the weights of tailored networks for the initialization. Diehl et al. (2015) proposed two ways to perform the weight normalization on the converted SNNs. The first one is called a model-based normalization that considers all possible positive activations and re-scale all the weights by the maximum positive input, and the second is called a data-based normalization in which the weights are normalized according to the maximum activation reached by propagating all the training examples through the network.

The numbers reported in Table 1 show that the SNNs trained with the proposed method outperform all the SNN baselines across 6 text classification datasets. They also achieved comparable results to the original TextCNNs by a small drop of $2.51\%$ on average in accuracy ($2.89\%$ difference for English and $1.78\%$ for Chinese respectively). The fine-tuned SNNs achieved up to $1.32\%$ improvement in accuracy ($0.61\%$ increase on average). Besides, the standard deviation decreased to 0.33 from 0.65 (almost halved) after the fine-tuning. We tried some combinations of "conversion + normalization + fine-tuning" and found that when the conversion is followed by the fine-tuning step, the weight normalization contributes a little to the performance of SNNs on the text classification tasks.

## 4.4 ADVERSARIAL ROBUSTNESS

Deep neural networks have proven to be vulnerable to adversarial examples (Samanta & Mehta, 2017; Wong, 2017; Liang et al., 2018; Alzantot et al., 2018), and the existence and pervasiveness of adversarial examples have raised serious concerns. We believe that SNNs provide a promising means to defend against adversarial attacks due to the non-differentiability of spikes and their tolerance to noise introduced by randomly-generated input spike trains. We evaluated the empirical robustness of

Table 2: Empirical results on 4 English datasets under 4 different adversarial attack algorithms on randomly selected $1,000$ examples for each dataset.

| Dataset | Model | Cln | TextFooler | | BERT-Attack | | TextBugger | | PWWS | |
|---------|-------|-----|------|------|------|------|------|------|------|------|
| | | | **Boa** | **Suc** | **Boa** | **Suc** | **Boa** | **Suc** | **Boa** | **Suc** |
| **MR** | TextCNN | 77.50 | 8.60 | 88.11 | 8.30 | 88.38 | 13.30 | 81.58 | 5.30 | 92.79 |
| | SNN | 74.30 | **16.40** | 77.47 | **10.40** | 85.91 | **22.70** | 69.45 | **12.20** | 82.98 |
| **SST-2** | TextCNN | 81.80 | 8.30 | 89.51 | 4.70 | 94.08 | 13.30 | 83.54 | 4.10 | 94.75 |
| | SNN | 80.20 | **14.70** | 81.39 | **9.20** | 88.22 | **21.50** | 72.37 | **11.00** | 86.11 |
| **Subj** | TextCNN | 93.50 | 11.10 | 87.12 | 7.40 | 91.28 | 12.30 | 84.60 | 6.40 | 92.31 |
| | SNN | 90.40 | **47.40** | 47.09 | **39.50** | 56.16 | **51.00** | 42.50 | **41.40** | 54.41 |
| **SST-5** | TextCNN | 44.80 | 0.70 | 98.31 | 0.30 | 99.33 | 1.80 | 95.74 | 0.50 | 98.80 |
| | SNN | 41.10 | **7.00** | 84.49 | **5.10** | 89.90 | **8.30** | 81.33 | **5.50** | 88.51 |

SNNs under test-time attacks with four black-box, synonym substitution-based attacks: TextFooler (Jin et al., 2020), TextBugger (Li et al., 2018a), BERT-Attack (Li et al., 2020), and PWWS (Ren et al., 2019). BERT-Attack generates synonyms dynamically by using BERT (Devlin et al., 2019), and all the other attack algorithms use $K$ nearest neighbor words of GloVe vectors (Pennington et al., 2014) to generate the synonyms of a word. In this experiment, we set the maximum percentage of words that can be modified to $0.3$, the size of the synonym set to $15$, and the semantic similarity threshold between an original text and the adversarial one to $0.8$. The following metrics (Li et al., 2021) are used to report the results of empirical robustness:

- The *clean accuracy* (**Cln**) is the accuracy achieved by a classifier on the clean texts.
- The *robust accuracy* (**Boa**) is the accuracy of a classifier achieved under a certain attack.
- The *success rate* (**Suc**) is the number of texts successfully perturbed by an attack algorithm (causing the model to make errors) divided by all the number of texts to be attempted.

Table 2 shows the clean accuracy, robust accuracy (i.e., accuracy under attack), and attack success rate achieved by the SNNs under four sophisticated attacks on all 4 English datasets, compared to the original TextCNNs. Following the evaluation setting used in (Li et al., 2021; Wang et al., 2021; Zhang et al., 2021), we randomly sampled $1,000$ examples from each test set to evaluate the models' adversarial robustness because it is prohibitively slow to attack the entire test set. For fair comparison, the ensemble method (see Subsection 4.2 for details) was not used in the SNNs when they were evaluated under adversarial attacks since existing studies show that ensemble methods can be used to improve the adversarial robustness (Strauss et al., 2017; Yuan et al., 2021). Therefore, the clean accuracy of SNNs reported in Table 2 is slightly lower than those in Table 1. From these numbers, we can see that the SNNs consistently perform better than the original neural networks under all four attack algorithms in both the robust accuracy and attack success rate while suffering little performance drop on the clean data. The SNNs can improve the robust accuracy under attack by average $13.55\%$ and lower the attack success rate by $17\%$ on average. On the Subj dataset, the robust accuracy can even be increased by a fairly significant margin of $38.7\%$ with $42.1\%$ decrease in the attack success rate under the test-time BERT-Attack.

### 4.5 ABLATION STUDY AND IMPACT OF HYPER-PARAMETERS

We conducted an ablation study over all the considered datasets on the SNNs obtained by two variants of training methods to analyze the necessity of the shallow training (i.e., conversion step) and the pre-trained word embeddings. One is to train SNNs directly with the surrogate gradients without the conversion step. As we can see from the row indicated by "Directly-trained SNN" from Table 1, the SNNs trained directly perform considerably worse than those trained with the two-step method by a significant margin of $21.17\%$ accuracy on average. It shows that the conversion step is indispensable in training SNNs. Another is to use randomly-generated word embeddings to train SNNs. We report these results in Table 3, where the models indicated by "RWE" were initialized with the word embeddings randomly generated, while those by "PRE" were with pre-trained word embeddings. No matter what training method is used, a significant drop in accuracy is observed in all the SNNs. Comparing the numbers shown in the last two rows of Table 3, we noticed a considerable difference of up to $1.39\%$ in accuracy on average between the SNNs with or without using pre-trained word embeddings, indicating their effectiveness in improving the performance of SNNs.

We want to understand how the performance and estimated energy consumption of SNNs are impacted by the choice of three important hyper-parameters: the number of neurons per category, the value

Table 3: Classification accuracy achieved by the models using randomly-generated word embeddings (denoted as "RWE") and those using pre-trained word embeddings (denoted as "PWE").

| Method | English Dataset | | | | Chinese Dataset | |
|---|---|---|---|---|---|---|
| | MR | SST-2 | Subj | SST-5 | ChnSenti | Waimai |
| Original TextCNN (RWE) | $75.02\pm0.19$ | $81.93\pm0.20$ | $92.20\pm0.23$ | $44.29\pm0.15$ | $84.53\pm0.18$ | $86.85\pm0.16$ |
| Tailored TextCNN (RWE) | $74.32\pm0.24$ | $81.59\pm0.17$ | $91.40\pm0.22$ | $42.41\pm0.18$ | $83.55\pm0.16$ | $86.79\pm0.14$ |
| Conv SNN (RWE) | $73.29\pm1.01$ | $79.10\pm0.83$ | $90.20\pm0.40$ | $39.81\pm0.69$ | $82.86\pm0.68$ | $86.01\pm0.38$ |
| Conv SNN + MN (RWE) | $73.51\pm0.54$ | $76.91\pm0.87$ | $89.20\pm0.47$ | $38.34\pm0.86$ | $82.97\pm0.51$ | $84.74\pm0.74$ |
| Conv SNN + DN (RWE) | $72.78\pm0.91$ | $79.57\pm0.95$ | $89.70\pm0.72$ | $38.47\pm1.14$ | $82.19\pm0.49$ | $85.94\pm0.53$ |
| Conv SNN + FT (RWE) | $74.06\pm0.42$ | $80.21\pm0.35$ | $90.30\pm0.26$ | $40.42\pm0.30$ | $83.75\pm0.07$ | $86.13\pm0.24$ |
| Conv SNN + FT (PWE) | $\mathbf{75.45}\pm0.51$ | $\mathbf{80.91}\pm0.34$ | $\mathbf{90.60}\pm0.32$ | $41.63\pm0.44$ | $\mathbf{85.02}\pm0.22$ | $\mathbf{86.66}\pm0.17$ |

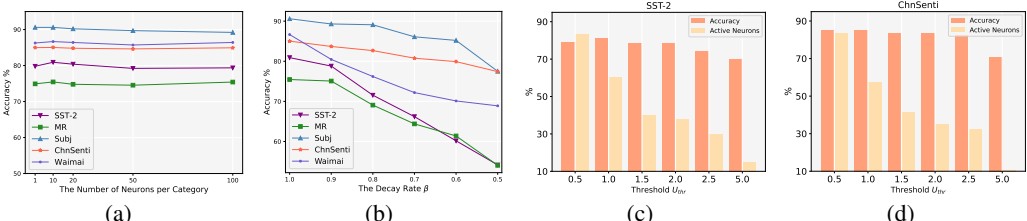

(a)     (b)     (c)     (d)

Figure 3: The impact of hyper-parameters. (a) Accuracy versus the number of neurons used per category. (b) Accuracy versus the decay rate of $\beta$. (c) and (d) Accuracy and the proportion of active neurons influenced by different values of membrane thresholds $U_{\text{thr}}$ on SST-2 and ChnSenti datasets.

of decay rate $\beta$, and the membrane threshold $U_{\text{thr}}$. As we mentioned in Subsection 4.2, more than one spiking neuron can be assigned to each category to improve the accuracy of prediction. Figure 3 (a) shows that the classification accuracy is generally insensitive to the number of neurons used for prediction, and the highest accuracy can be achieved around 10 neurons per category. It appears that if more neurons are used, the SNNs suffer from the problem of over-fitting. As we can see from Figure 3 (b), if the conversion-based method is used the value of $\beta$ should be set to 1.0, otherwise the accuracy will be severely degraded. A single spike only consumes a constant amount of energy (Cao et al., 2015). The amount of energy consumption heavily depends on the number of spikes and the number of time steps used at the inference stage. Figure 3 (c) and (d) show that we can reduce the number of active neurons (approximately the number of spikes) by increasing the value of membrane threshold while suffering little or no performance drop, which implies the possibility of about 50% energy saving by carefully adjusting the values of membrane threshold (say $U_{\text{thr}} = 2$). In Appendix A.4, we show that the SNNs can reduce more than 10 times the energy consumption on average, compared to conventional TextCNNs. We also want to understand how the choice of the number of time steps impacts the accuracy of SNNs (see Appendix A.5 for details). We found that the fine-tuned SNNs using 50 time-steps outperform all the converted SNNs without the fine-tuning including those using 80 time-steps, indicating that the proposed fine-tuning method can significantly speed up the inference time and reduce the energy consumption while maintaining the accuracy.

## 5 CONCLUSION

We found that it is hard to train spiking neural networks for language tasks directly using the error backpropagation through time although SNNs are supposed to be suitable for modeling time-varying data due to their temporal dynamics. To address this issue, we suggested a two-step training recipe: start with an SNN converted from a normally-trained tailored network, and perform backpropagation on the converted SNN. We also proposed a method to make use of pre-trained word embeddings in SNNs. Pre-trained word embeddings are projected into vectors with positive values after proper normalization and shifting, which can be used to initialize tailored networks and converted to spike trains as input of SNNs. Through extensive experimentation on 6 text classification datasets, we demonstrated that the SNNs trained with the proposed method achieved competitive results on both English and Chinese datasets. Such SNNs were also proven to be less vulnerable to textual adversarial examples than traditional neural counterparts. It would be interesting to see if we can pre-train SNNs unsupervisedly using masked language modeling with a large collection of text data in the future.

## ACKNOWLEDGMENTS

The authors would like to thank the anonymous reviewers for their valuable comments. This work was partly supported by National Natural Science Foundation of China (No. 62076068), Shanghai Municipal Science and Technology Major Project (No. 2021SHZDZX0103), and Shanghai Municipal Science and Technology Project (No. 21511102800).

## REPRODUCIBILITY STATEMENT

The authors have made great efforts to ensure the reproducibility of the empirical results reported in this paper. Firstly, the experiment settings, evaluation metrics, and datasets were described in detail in Subsections 4.1, 4.4, and 4.5. Secondly, the implementation details were clearly presented in Subsections 4.2 and Appendix A.1. Finally, the source code is avaliable at `https://github.com/Lvchangze/snn`.

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

## A APPENDIX

### A.1 THE DERIVATIVE OF THE LOSS WITH RESPECT TO THE WEIGHTS

Given a loss function defined in Equation (4), the losses at every time step can be summed together to give the following global gradient, as illustrated in Figure 2 (b):

$$\frac{\partial L}{\partial W} = \sum_t \frac{\partial L_t}{\partial W} = \sum_i \sum_{j \leq i} \frac{\partial L_i}{\partial W_j} \frac{\partial W_j}{\partial W} \tag{5}$$

where $i$ and $j$ denote different time steps, and $L_t$ is the loss calculated at the step $t$. No matter which time step is, the weights of an SNN are shared across all steps. Therefore, we have $W_0 = W_1 = \cdots = W$, which also indicates that $\frac{\partial W_j}{\partial W} = 1$. Thus, Equation (5) can be written as follows:

$$\frac{\partial L}{\partial W} = \sum_i \sum_{j \leq i} \frac{\partial L_i}{\partial W_j} \tag{6}$$

Based on the chain rule of derivatives, we obtain:

$$\begin{aligned}
\frac{\partial L}{\partial W} &= \sum_i \sum_{j \leq i} \frac{\partial L_i}{\partial S_i} \frac{\partial S_i}{\partial U_i} \frac{\partial U_i}{\partial W_j} \\
&= \sum_i \frac{\partial L_i}{\partial S_i} \frac{\partial S_i}{\partial U_i} \sum_{j \leq i} \frac{\partial U_i}{\partial W_j}
\end{aligned} \tag{7}$$

where $\frac{\partial L_i}{\partial S_i}$ is the derivative of the cross-entropy loss at the time step $i$ with respect to $S_i$, and $\frac{\partial S_i}{\partial U_i}$ can be easily derived from Equation (3). As to the last term of $\sum_{j \leq i} \frac{\partial U_i}{\partial W_j}$, we can split it into two parts:

$$\sum_{j \leq i} \frac{\partial U_i}{\partial W_j} = \frac{\partial U_i}{\partial W_i} + \sum_{j \leq i-1} \frac{\partial U_i}{\partial W_j} \tag{8}$$

From Equation (2), we know that $\frac{\partial U_i}{\partial W_i} = X_i$. Therefore, Equation (5) can be simplified as follows:

$$\frac{\partial L}{\partial W} = \sum_i \underbrace{\frac{\partial L_i}{\partial S_i} \frac{\partial S_i}{\partial U_i}}_{\text{constant}} \left( \underbrace{\frac{\partial U_i}{\partial W_j}}_{\text{constant}} + \sum_{j \leq i-1} \frac{\partial U_i}{\partial W_j} \right) \tag{9}$$

By the chain rule of derivatives over time, $\frac{\partial U_i}{\partial W_j}$ can be factorized into two parts:

$$\frac{\partial U_i}{\partial W_j} = \frac{\partial U_i}{\partial U_{i-1}} \frac{\partial U_{i-1}}{\partial W_j} \tag{10}$$

It is easy to see that $\frac{\partial U_i}{\partial U_{i-1}}$ is equal to $\beta$ from Equation (2), and Equation (5) can be written as:

$$\frac{\partial L}{\partial W} = \sum_i \underbrace{\frac{\partial L_i}{\partial S_i} \frac{\partial S_i}{\partial U_i}}_{\text{constant}} \left( \underbrace{\frac{\partial U_i}{\partial W_j}}_{\text{constant}} + \sum_{j \leq i-1} \underbrace{\frac{\partial U_i}{\partial U_{i-1}}}_{\text{constant}} \frac{\partial U_{i-1}}{\partial W_j} \right) \tag{11}$$

We can treat $\frac{\partial U_{i-1}}{\partial W_j}$ recurrently as Equation (8). Finally, we can update the weights $W$ by the rule of $W = W - \eta \frac{\partial L}{\partial W}$, where $\eta$ is a learning rate.

### A.2 DATASET

- **MR**: It consists of movie-review documents labeled with respect to their overall sentiment polarity (positive or negative) or subjective rating (Pang & Lee, 2005).
- **SST-5**: The Stanford Sentiment Treebank 5 comprises $11,855$ sentences extracted from movie reviews for sentiment classification (Socher et al., 2013). There are $5$ different classes (very negative, negative, neutral, positive, and very positive).

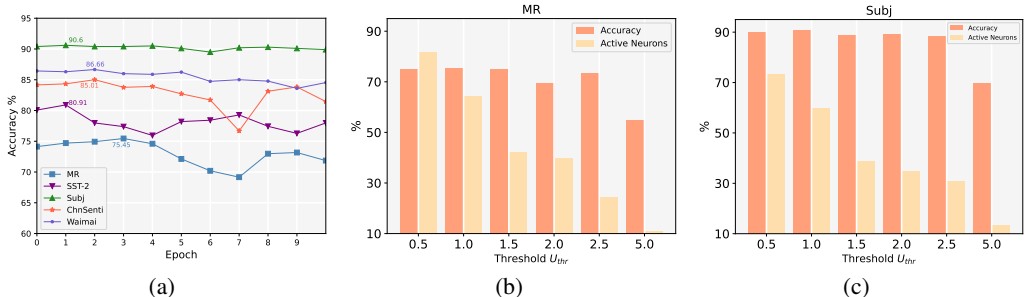

(a)                                   (b)                                   (c)

Figure 4: (a) Classification accuracy versus the number of epochs used to fine-tune SNNs (b) and (c) Accuracy and the proportions of active neurons influenced by different values of membrane thresholds $U_{\text{thr}}$ on MR and Subj datasets.

- **SST-2**: It is the binary version of SST-5, and there are just 2 classes (positive and negative).
- **Subj**: The task of this dataset is to classify a sentence as being subjective or objective[1].
- **ChnSenti**: This datasets contains about $7,000$ Chinese hotel reviews annotated with positive or negative labels[2].
- **Waimai**: It consists of $12,000$ Chinese user reviews collected by a food delivery platform for binary sentiment classification (positive and negative)[3].

## A.3    MORE ANALYSIS ON THE IMPACT OF HYPER-PARAMETERS

Figure 4 (a) shows how the accuracy of SNNs varies as the number of epochs grows at the fine-tuning phase. Although the highest accuracy is achieved with different numbers of epochs for different datasets, the peak accuracy is always reached within $5$ epochs, indicating that just a few epochs are required to fine-tune the converted SNNs and such additional training time and effort are acceptable. For each value of membrane threshold $U_{\text{thr}}$, the corresponding accuracy and the proportion of active spiking neurons on both MR and Subj datasets are reported in Figure 4 (b) and (c) respectively. We found similar trends as those for SST-2 and ChnSenti datasets shown in Figure 3 (c) and (d), which confirms that the energy consumption can be further reduced by about $50\%$ without suffering much performance loss. We also investigate the impact of the dropout technique on the performance of the resulting SNNs. Table 4 shows the accuracy on $6$ text classification datasets achieved by different neural models that were trained without using the dropout technique (Srivastava et al., 2014). By comparing the numbers reported in Tables 1 (where the dropout rate was set to $50\%$ during the training process) and 4, it is clear that the dropout technique is still quite useful to train spike neural networks although its contribution to the performance of SNNs is slightly smaller than that of DNNs.

## A.4    COMPARISON OF ENERGY CONSUMPTION

We compare theoretical energy consumption of TextCNNs and SNNs on $6$ different text classification test datasets, and reported the results in Table 5. The way to calculate the number of floating point operations (FLOPs), the number of synaptic operations (SOPs), and the average theoretical energy consumption (Power) will be discussed later. As we can see the numbers from Table 5, classical TextCNNs demand more than $10$ times the energy consumption on average, compared to SNNs. On the test set of Waimai, the SNN can reduce up to $14.4$ times (i.e., $93.05\%$ decrease) average energy consumption required for predicting each text example, compared to the corresponding TextCNN.

For spiking neural networks (SNNs), the theoretical energy consumption of layer $\xi$ can be calculated as $\text{Power}(\xi) = 77\text{fJ} \times \text{SOPs}(\xi)$, where $77\text{fJ}$ is the energy consumption per synaptic operation (SOP) (Indiveri et al., 2015; Hu et al., 2018). The number of synaptic operations at the layer $\xi$ of an SNN is estimated as $\text{SOPs}(\xi) = T \times \gamma \times \text{FLOPs}(\xi)$, where $T$ is the number of times step required in the simulation, $\gamma$ is the firing rate of input spike train of the layer $\xi$, and $\text{FLOPs}(\xi)$ is the estimated floating

---

[1] https://www.cs.cornell.edu/people/pabo/movie-review-data/

[2] https://raw.githubusercontent.com/SophonPlus/ChineseNlpCorpus/master/datasets/ChnSentiCorp_htl_all/ChnSentiCorp_htl_all.csv

[3] https://raw.githubusercontent.com/SophonPlus/ChineseNlpCorpus/master/datasets/waimai_10k/waimai_10k.csv

Table 4: Classification accuracy achieved by different models on 6 datasets. These models were trained without using the dropout technique. The model obtained by applying the model-based normalization on the converted SNN is denoted as "Conv SNN + MN" and that by applying the data-based normalization as "Conv SNN + DN". The SNNs trained with the "conversion + fine-tuning" is denoted as "Conv SNN + FT".

| Method | English Dataset | | | | Chinese Dataset | |
|---|---|---|---|---|---|---|
| | MR | SST-2 | Subj | SST-5 | ChnSenti | Waimai |
| Original TextCNN | $76.29 \pm 0.25$ | $82.70 \pm 0.18$ | $92.60 \pm 0.20$ | $43.40 \pm 0.23$ | $85.11 \pm 0.14$ | $87.82 \pm 0.17$ |
| Tailored TextCNN | $75.91 \pm 0.19$ | $82.26 \pm 0.23$ | $92.50 \pm 0.23$ | $42.67 \pm 0.25$ | $84.43 \pm 0.15$ | $87.82 \pm 0.18$ |
| Conv SNN | $74.68 \pm 1.16$ | $73.41 \pm 0.95$ | $86.50 \pm 0.42$ | $35.29 \pm 1.20$ | $69.61 \pm 0.64$ | $84.63 \pm 0.53$ |
| Conv SNN + MN | $57.83 \pm 1.51$ | $73.75 \pm 0.74$ | $89.50 \pm 0.54$ | $29.10 \pm 0.95$ | $68.93 \pm 0.73$ | $80.46 \pm 0.64$ |
| Conv SNN + DN | $74.73 \pm 0.54$ | $75.38 \pm 0.53$ | $87.00 \pm 0.65$ | $35.48 \pm 1.03$ | $70.45 \pm 0.57$ | $84.35 \pm 0.43$ |
| Conv SNN + FT | $\mathbf{75.16} \pm 0.52$ | $\mathbf{75.45} \pm 0.30$ | $\mathbf{90.50} \pm 0.34$ | $\mathbf{38.87} \pm 0.41$ | $\mathbf{83.99} \pm 0.28$ | $\mathbf{86.04} \pm 0.25$ |

Table 5: Comparison of energy consumption on 6 text classification benchmarks. The floating point operations of TextCNN are denoted as "FLOPs" and the synaptic operations of SNNs as "SOPs". The average theoretical energy required for each test example prediction is indicated by "Power".

| Dataset | Model | FLOPs / SOPs(G) | Power (mJ) | Energy Reduction | Accuracy (%) |
|---|---|---|---|---|---|
| MR | TextCNN | 0.36 | 4.498 | $\mathbf{10.66 \times \downarrow}$ | 77.41 |
| | SNN | 5.49 | 0.422 | | 75.45 |
| SST-2 | TextCNN | 0.25 | 3.140 | $\mathbf{9.05 \times \downarrow}$ | 83.25 |
| | SNN | 4.51 | 0.347 | | 80.91 |
| Subj | TextCNN | 0.36 | 4.478 | $\mathbf{9.59 \times \downarrow}$ | 94.00 |
| | SNN | 6.06 | 0.467 | | 90.60 |
| SST-5 | TextCNN | 0.25 | 3.108 | $\mathbf{9.14 \times \downarrow}$ | 45.48 |
| | SNN | 4.41 | 0.340 | | 41.63 |
| ChnSenti | TextCNN | 0.33 | 4.144 | $\mathbf{7.31 \times \downarrow}$ | 86.74 |
| | SNN | 7.37 | 0.567 | | 85.02 |
| Waimai | TextCNN | 0.33 | 4.132 | $\mathbf{14.40 \times \downarrow}$ | 88.49 |
| | SNN | 3.72 | 0.287 | | 86.66 |

point operations at the layer $\xi$. For classical artificial neural networks like TextCNNs, the theoretical energy consumption required by the layer $\xi$ can be estimated by $\text{Power}(\xi) = 12.5\text{pJ} * \text{FLOPs}(\xi)$. Note that $1\text{J} = 10^3 \text{ mJ} = 10^{12} \text{ pJ} = 10^{15} \text{ fJ}$.

## A.5    THE IMPACT OF THE NUMBER OF TIME STEPS

We want to understand how the choice of the number of time steps used at the inference stage impacts the accuracy of SNNs by varying the number of times steps from 10 to 80. As we can see Figure 5 (a), generally the larger the number of time steps the higher the accuracy achieved by the SNNs. However, the performance increases smoothly when the number of time steps is larger than 50. It is noteworthy that the inference time of SNNs that are converted from ANNs for computer vision tasks turns out be very large (of the order of a few thousand-time steps).

We also would like to know whether the fine-tuning can help to reduce the number of time steps required to achieve reasonable performance. In Figure 5 (b) and (c), we report accuracy achieved by the SNNs with and without the fine-tuning on English and Chinese text classification benchmarks respectively. We found that on all the considered datasets the fine-tuned SNNs with 50 time-steps outperform the converted SNNs (without the fine-tuning) using any time step between 20 and 80 at the inference, even those with the largest 80 time-steps. On the Subj, ChnSenti, and Waimai datasets, the fine-tuned SNNs using 40 time-steps can beat all the converted SNNs without the fine-tuning phase including those using 80 time-steps, indicating that the proposed fine-tuning method can significantly speed up the inference time and reduce the energy consumption while maintaining the accuracy.

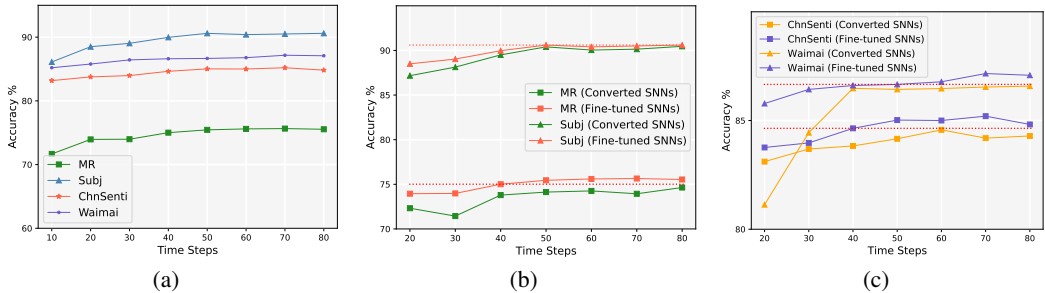

Figure 5: Classification accuracy versus the number of time steps. (a) The accuracy achieved by the fine-tuned SNNs with various time steps at the inference time on the test sets of MR, Subj, ChnSenti, and Waimai datasets. (b) The accuracy achieved by the SNNs with and without the fine-tuning on two English text classification benchmarks. (c) The accuracy achieved by the SNNs with and without the fine-tuning on two Chinese text classification datasets.

## A.6 MOTIVATION AND LIMITATIONS

Unlike classical artificial neural networks (ANNs), spiking neural networks (SNNs) do not transmit information in form of continuous values, but rather the time when a membrane potential reaches a specific threshold. Once the membrane potential reaches the threshold, the neuron fires and generates a pulse signal that travels to the downstream neurons which increase or decrease their potentials in proportion to the connection strengths in response to this signal. SNNs incorporate the concept of time into their computing model in addition to neuronal and synaptic states. They are considered to be more biologically plausible neuronal models than classical ANNs. Besides, SNNs are suitable for implementation on low power hardware, and offer a promising computing paradigm to deal with large volumes of data using spike trains for information representation in a more energy-efficient manner. Nowadays, excessive energy consumption is a major impairment to more wide-spread applications of ANNs. Spike-based neuromorphic hardware now are available to alleviate this problem by more energy-efficient implementations of ANNs than specialized hardware such as GPUs. It has been reported that improvements in energy consumption of up to $2 \sim 3$ orders of magnitude when compared to conventional ANN acceleration on embedded hardware (Azghadi et al., 2020; Ceolini et al., 2020; Davies et al., 2021). For more introduction to SNNs, we refer readers to several good reviews (Roy et al., 2019; Tavanaei et al., 2019; Taherkhani et al., 2020; Eshraghian et al., 2021).

Many neuromorphic systems now allow us to simulate software-trained models without performance loss. Since mature on-chip training solutions are not yet available, it remains a great challenge to deploy high-performing SNNs on such hardware due to the lack of efficient training algorithms. In addition, there have been very few works that have demonstrated the efficacy of SNNs in natural language processing (NLP) tasks. This study shows how encoding pre-trained word embeddings as spike trains and training with the two-step recipe (conversion + fine-tuning) can yield competitive performance on multiple text classification benchmarks both for English and Chinese languages, thereby giving us a glimpse of how learning algorithms can empower neuromorphic technologies for energy-efficient and ultralow-latency language processing in the future. SNNs still lag behind ANNs in terms of accuracy yet. Through intensive research on SNNs in recent years, the performance gap between deep neural networks (DNNs) and SNNs is constantly narrowing, and can even vanish on some vision tasks. SNNs cannot currently outperform DNNs on the datasets that were created to train and evaluate conventional DNNs (they use continuous values). Such data should be converted into spike trains before it can be feed into SNNs, and this conversion might cause loss of information and result in a reduction in performance. Therefore, the comparison is indirect and unfair. New datasets that have properties which are compatible with SNNs are expected to be available in the near future, such as those obtained by event-based cameras (Ramesh et al., 2019) or the spiking activities that are recorded from biological nervous systems (Maggi et al., 2018), which could be difficult for classical DNNs. For language processing, it would be interesting to see if we can pre-train SNNs unsupervisedly using masked language modeling with a large collection of text data and narrow the performance gap between SNNs and start-of-the-art transformer-based models in the future.

