# OpenReview forum: "Spiking Convolutional Neural Networks for Text Classification"
_ICLR.cc/2023/Conference — ICLR 2023 poster_

### Official Review · Reviewer_xeVA · 2022-10-22

**Confidence:** 5
**Correctness:** 4
**Technical Novelty And Significance:** 3
**Empirical Novelty And Significance:** 4
**Recommendation:** 6

**Clarity, Quality, Novelty And Reproducibility:**

The idea is clear and the paper is well written and organized. The idea is not very novel but the authors applied it in a new application.

**Strength And Weaknesses:**

Strength:
1. There are few works that demonstrate the efficacy of SNNs in language tasks. This paper makes a breakthrough.
2. The results are excellent. The proposed models achieve comparable results to the original TextCNNs.
3. The idea is clear and the paper is well organized.

Weakness:
1. The idea is not very novel. The whole pipeline is quite similar to the converting process of original SCNNs in the image recognition domain.
2. The spiking encoding may lead to the increasing of time steps during inference and training for each word embeddings that are converted to spike trains periodically. The authors did not mention that.

**Summary Of The Paper:**

This work presents a “conversion +fine-tuning” two-step method for training SNNs for text classification. Meanwhile, a new encoding method for converting pre-trained word embeddings to the spiking version is proposed. As results, the converted SNNs achieved comparable results compared with DNNs on text classification benchmarks. Also, the proposed model shows better robustness against adversarial attack than DNNs.

**Summary Of The Review:**

This work fills the gap in the field of SNNs for text classification tasks. The experimental results are excellent and the conclusions are well supported. But the authors need to justify the impact of time steps during inference and training of the converted SNNs.

---

> ### Author Response · Authors · 2022-11-13
> **Response to Reviewer xeVA.**
>
> Thank you for your valuable comments.
>
> Q1: the authors need to justify the impact of time steps during inference and training of the converted SNNs.
>
> R1: It is a very good suggestion, and we have conducted a set of experiments to understand the how the choice of the number of time steps impacts the accuracy of SNNs (see Appendix A.4). As we can see Figure 5 (a), generally the larger the number of time steps the higher the accuracy achieved by the SNNs. However, the performance increases smoothly when the number of time steps is larger than 50. It is noteworthy that the inference time of SNNs that are converted from ANNs for computer vision tasks turns out be very large (of the order of a few thousand-time steps). We also would like to know whether the fine-tuning can help to reduce the number of time steps required to achieve reasonable performance. In Figure 5 (b) and (c), we report accuracy achieved by the SNNs with and without the fine-tuning on English and Chinese text classification benchmarks respectively. We found that on all the considered datasets the fine-tuned SNNs with 50 time-steps outperform the converted SNNs (without the fine-tuning) using any time step between 20 and 80 at the inference, even those with the largest 80 time-steps. On the Subj, ChnSenti, and Waimai datasets, the fine-tuned SNNs using 40 time-steps can beat all the converted SNNs without the fine-tuning stage including those using 80 time-steps, indicating that the proposed fine-tuning method can significantly speed up the inference time and reduce the energy consumption while maintaining the accuracy.

---

### Official Review · Reviewer_SuuR · 2022-10-25

**Confidence:** 3
**Correctness:** 4
**Technical Novelty And Significance:** 4
**Empirical Novelty And Significance:** 4
**Recommendation:** 8

**Clarity, Quality, Novelty And Reproducibility:**

The paper is very well-written and organized, and is a compelling read.

The research program and experiments are well-scoped and well-thought out.

The findings extend the current art meaningfully.

The Introduction lays out relevant prior work. I do not have sufficient background in the literature to confirm completeness of citations or novelty (so the scores I give for Novelty might be wrong).

**Strength And Weaknesses:**


(strengths)

Well-written and organized.

Extends viability of SNNs to large vocabularies.

A wide array of example tasks, in 2 languages.

(weaknesses)

The opening motivation, reducing energy consumption, is less impactful than it appears because the proposed method requires pre-training and fine-tuning a standard ANN before conversion to an SNN. So the only savings are over inference time. For a very few successful and widely deployed algorithms this will be significant, but for most algorithms during research, there is little difference in energy cost. This distinction should perhaps be clarified.

(Miscellaneous)

Table 1: is it possible to report mean +/- std dev over k-fold splits? This gives a much more powerful and informative report.

Table 2: Since the SNNs are still clearly harmed by the adversarial attacks, perhaps you could discuss whether there are ways to improve on this (or conversely comment that robustness, while much better than CNNs, is unlikely to improve).

Page 1: is the "question classification task" one of the 17 toy tasks, or is it substantially harder?

Page 1: "which is less biologically plausible": Does this matter at all? perhaps this clause could be removed.

Bottom page 2: Maybe replace "former" and "latter" with "conversion" and "spike-based" for clarity.

Section 3.1: "is trained by the same way as one would with " -> "is trained in the same way as"

Section 3.1: " some operations that cannot be realized by ": this raises an instant question in the reader "what operations?" which is not answered until 3.1.1. Perhaps add a pointer to that future list, eg "(listed in 3.1.1)"

3.1.1: "method is applied" -> method can be applied

Top of page 4: "embeddings learn" -> "embeddings to learn"

3.1.1 third bullet: It seems that removing biases would affect whether units activated or not. Why does removing biases not matter?

"clamp": maybe "clip" is more standard?

"dividing by sigma": divide by 6*sigma?

Figure 2: This should be moved to after eqn 2, so that the reader knows what the variables mean. In its current location, the figure is not comprehensible when encountered.

"illustrated in Figure 2" -> "2a"

3.2.3: It looks like the target vector increases as the vocabulary size. How big is it, and are there limits to vocabulary size due to this part of the architecture?

"3.2.3: "we list the efficient" -> "an"




**Summary Of The Paper:**

The paper applies SNNs to text dataset tasks. It proposes a method of converting a pre-trained standard ANN into an SNN. This improves on current methods by allowing the model to handle large vocabularies.

**Summary Of The Review:**

The paper documents a well-designed research program and experiments that extend current art in the field of SNNs.

The paper itself is well-written and a pleasure to read.

---

> ### Author Response · Authors · 2022-11-13
> **Response to Reviewer SuuR.**
>
> Thank you for your insightful comments.
>
> Q1: The opening motivation, reducing energy consumption, is less impactful than it appears because the proposed method requires pre-training and fine-tuning a standard ANN before conversion to an SNN. So, the only savings are over inference time.
>
> R1: Yes, the energy consumption is mainly reduced at the inference time. Once SNNs are well software-trained, they can be deployed on neuromorphic hardware for energy-efficient computing. However, mature on-chip training solutions are not yet available, and it remains a great challenge due to the lack of efficient training algorithms, even in a software training environment. Thank you for pointing it out, and we have made it clear in the revised version. We also have added a comparison of synaptic operations and theoretical energy consumption (see Appendix A.3). We found that conventional TextCNNs demand more than 10 times the energy consumption on average, compared to SNNs.
>
> Q2: Table 1: is it possible to report mean +/- std dev over k-fold splits?
>
> R2: In all the experiments, we followed the standard training and test split defined for each dataset (if any). All the reported average accuracies were obtained over 10 runs with different random initialization for each setting. Following your suggestion, we have reported the standard deviation in Tables 1, 3, and 4. From those numbers, we also found that the standard deviation decreased to 0.33 from 0.65 (almost halved) after the fine-tuning, compared to the simply converted SNNs. The fine-tuning also helps to reduce the number of time steps required to achieve reasonable performance (see Appendix A.4), which shows that the computational cost invested at the fine-tuning stage is worth it.
>
> Q3: Since the SNNs are still clearly harmed by the adversarial attacks, perhaps you could discuss whether there are ways to improve on this.
>
> R3: Like classical ANNs, there are many methods that can be used to improve the adversarial robustness of SNNs. For example, adversarial data augmentation is one of the widely used methods [1][2]. During the training phase, they replace a word with one of its synonyms that maximizes the prediction loss. By augmenting these adversarial examples with the original training data, the models are trained to be robust to such perturbations. In this study, we would like to point out that SNNs are inherently far more robust than ANNs because it is harder for any adversaries to break SNNs due to the non-differentiability of spikes, which hinders the adversaries to peek the changes in the model's predictions (or gradients) by a small perturbation on the input. Besides, pathways that are susceptible to adversarial perturbations can learn to be represented as rate codes, which could alleviate timing perturbations in temporal codes.
>
> Q4: Is the "question classification task" one of the 17 toy tasks, or is it substantially harder?
>
> R4: We are not sure whether the question classification task is substantially harder than any one of 17 toy tasks. The question classification data set was presented in [3]. The goal of this task is not to answer the questions, but to classify question sentences into one of six categories. For example, the questions “How much does the human adult female brain weight?” expects an answer of the type “Number”.
>
> Q5: "which is less biologically plausible": Does this matter at all? perhaps this clause could be removed.
>
> R5: As a new generation of brain-inspired networks, many researchers in SNNs seek to find biologically plausible architectures and learning mechanisms to provide higher processing abilities. SNNs are more biologically realistic than ANNs, and arguably the only viable option if one wants to understand how the train computes. However, we agree with you on this, and this clause has been removed from the revised version. Thank you.
>
> Q6: Why does removing biases not matter?
>
> R6: SNNs are trained to optimize the performance regarding the cost function by updating the weights. A change to the weight causes a change of the membrane potential, which ultimately results in a change of spike timing. The effect of biases during computation is subsumed into the threshold dynamics of the neuron.
>
> Q7: It looks like the target vector increases as the vocabulary size. How big is it, and are there limits to vocabulary size?
>
> R7: The size of the target vectors is equal to the number of categories for classification. It does not increase as the vocabulary size. In Equation (4), T is used to denote the number of time steps, and N is the number of training instances.
>
> Reference:
>
> [1] Jin et al. Is BERT really robust? a strong baseline for natural language attack on text classification and entailment. AAAI2021.
>
> [2] Zheng et al. Evaluating and enhancing the robustness of neural network-based dependency parsing models with adversarial examples. ACL2020.
>
> [3] Li et al. Learning question classifiers. COLING2002.

---

> > ### Comment · Reviewer_SuuR · 2022-11-22
> > **Response to authors' comments**
> >
> > Thank you to the other reviewers, who saw and raised issues I had not considered.
> >
> > Thanks also to the authors for their itemized comments to each reviewer.
> >
> > I reiterate my support for accepting this paper, for a couple reasons:
> > 1. The reservations of other reviewers did not strike me as disqualifying (disclaimer: I am not versed in this literature).
> > 2. (relevant in particular to reviewer PFzn's comment) The scope of the paper is certainly incremental (it extends image-directed schemes to text), but I view this approach as a viable and effective way to advance the field.
> > Thank you.

---

### Official Review · Reviewer_PFzn · 2022-10-27

**Confidence:** 3
**Correctness:** 3
**Technical Novelty And Significance:** 2
**Empirical Novelty And Significance:** 2
**Recommendation:** 3

**Clarity, Quality, Novelty And Reproducibility:**

The paper is quite easy to follow. However, being spiking neural networks quite a niche field, I would have hoped the authors had introduced them in more details to help the reader understand the SNN setup and differences from regular neural networks (if not in the main body, at least in the appendix).

**Strength And Weaknesses:**

Strengths:
- Simple approach, easy to follow.

Weaknesses:
- Contributions are incremental and only marginally novel.
- This approach still requires training a regular TextCNN, partially defeating the motivation behind SNNs.
- Experimental section is quite limited and inconclusive. In particular, the lift from using fine-tuning vs simply using a converted SNN is rather marginal.
- Experimental robustness section: why are the "Clean" metrics in Table 2 different from the ones in Table 1. If that's because of the variance of results over different run, then the observed variance would invalidate the results and ranking presented in Table 1.

**Summary Of The Paper:**

The paper focuses on applying spiking neural networks (SNNs) to text classification tasks.

Spiking neural networks more closely mimic biological neural networks, with the main difference from typical artificial neural network being a temporal quality: each neuron accumulates a membrane charge over time (across inferences) and only transmits information when its threshold is hit. When this event happens, the neuron is said to fire (or spike), it resets itself, and its signal travels to the connected neurons which in response will increase or decrease their charge.

The main contribution is to convert and fine-tune a regular TextCNN architecture into its SNN counterpart. In particular,
(i) the given TextCNN is tailored to replace or remove unsupported operations such as word embeddings, negative values, biases, max-pooling.
(ii) This tailored NN is then trained with Gradient Descent
(iii) The corresponding SNN is initialized with the weights learned in the previous step, and then is fine-tuned using surrogate gradients.

Experiments are conducted on sentiment classification tasks.


**Summary Of The Review:**

Given the limited novelty and improvements, my recommendation is to reject this paper.

---

> ### Author Response · Authors · 2022-11-13
> **Response to Reviewer PFzn.**
>
> Thank you for your valuable comments.
>
> Q1: This approach still requires training a regular TextCNN, partially defeating the motivation behind SNNs.
>
> R1: Spiking neural networks (SNNs) can be deployed on neuromorphic hardware for energy-efficient computing once they are well trained. Most neuromorphic systems allow us to simulate software-trained models without performance loss. Since mature on-chip training solutions are not yet available, it remains a great challenge due to the lack of efficient training algorithms. The conversion-based technique is one of popular approaches to train SNNs. By this approach, the state-of-the-art methods for training classical artificial neural networks (ANNs) can be leveraged to build SNNs.
>
> Q2: The lift from using fine-tuning vs simply using a converted SNN is rather marginal.
>
> R2: It is well-known that SNNs are hard to train. The SNNs fine-tuned with the proposed method achieved up to 1.32% improvement in accuracy (+ 0.61% on average) on 6 different text classification benchmarks. Besides, the standard deviation decreased to 0.33 from 0.65 (almost halved) after the fine-tuning (see Table 1). We also found that the fine-tuned SNNs require much fewer time steps at the inference time than simply converted ones, which means far less latency and more energy-efficient afforded by the fine-tuned SNNs (see Appendix A.4).
>
> Q3: Why are the "Clean" metrics in Table 2 different from the ones in Table 1. If that's because of the variance of results over different run, then the observed variance would invalidate the results and ranking presented in Table 1.
>
> R3: For fair comparison, the ensemble method (i.e., multiple spiking neurons are used for each category and the prediction results on multiple neurons are ensembled to get a final output, see Subsection 4.2 for details) was not used in the SNNs when they were evaluated under adversarial attacks since existing studies show that ensemble methods can be used to improve the adversarial robustness [4][5]. The clean accuracies reported in Table 1 were obtained on all the test examples. Following the evaluation setting used in [1][2][3], we randomly sampled 1,000 examples from the test set to evaluate the models' adversarial robustness and reported the results in Table 2, because it is prohibitively slow to attack the entire test set. Therefore, the clean accuracies reported in Table 2 are slightly lower than those in Table 1. All the reported average accuracies were obtained over 10 runs with different random initialization for each setting of the network. Thank you so much for pointing it out, and we have made it clear in the revised version.
>
> Q4: I would have hoped the authors had introduced them in more details to help the reader understand the SNN setup and differences from regular neural networks (if not in the main body, at least in the appendix).
>
> R4: It is a good suggestion. We have added a brief introduction to SNNs in the Appendix A.5. Thank you.
>
> Reference:
>
> [1] Zongyi Li, Jianhan Xu, Jiehang Zeng, Linyang Li, Xiaoqing Zheng, Qi Zhang, Kai-Wei Chang, and Cho-Jui Hsieh. Searching for an effective defender: Benchmarking defense against adversarial word substitution. In Proceedings of the Conference on Empirical Methods in Natural Language Processing, 2021.
>
> [2] Xiaosen Wang, Jin Hao, Yichen Yang, and Kun He. Natural language adversarial defense through synonym encoding. In Proceedings of the Thirty- Seventh Conference on Uncertainty in Artificial Intelligence, 2021.
>
> [3] Yuhao Zhang, Aws Albarghouthi, and Loris D’Antoni. Certified robustness to programmable transformations in LSTMs. In Proceedings of the Conference on Empirical Methods in Natural Language Processing, 2021.
>
> [4] Thilo Strauss, Markus Hanselmann, Andrej Junginger and Holger Ulmer. Ensemble methods as a defense to adversarial perturbations against deep neural networks. ArXiv, abs/1709.03423, 2017.
>
> [5] Liping Yuan, Xiaoqing Zheng, Yi Zhou, Cho-Jui Hsiech, Kai-Wei Chang. On the transferability of adversarial attacks against neural text classifier. In Proceedings of Conference on Empirical Methods on Natural Language Processing, 2021.

---

### Official Review · Reviewer_fFGu · 2022-10-28

**Confidence:** 3
**Clarity, Quality, Novelty And Reproducibility:** 1. the writing of this paper is good.…
**Correctness:** 3
**Technical Novelty And Significance:** 3
**Empirical Novelty And Significance:** 3
**Recommendation:** 5

**Strength And Weaknesses:**

Strength
1. design SNN for text classification further extends the applications of SNN model;


Weaknesses
1. after a series of operations, the ANN is transformed into SNN version, however, the accuracy and running efficiency are all dropped? A natural question is do we have to conduct such a transformation?
2. the Transformer networks achieve remarkable performance in the NLP tasks, would people select the SNN for their application? I doubt this selection.
3. The experimental results are not impressive.

**Summary Of The Paper:**

This paper proposes a new scheme for training the SNN model to achieve text classification in the NLP community. Due to the fact that it is hard to directly train SNN for language tasks using the error backpropagation through time. The authors start with an SNN converted from a normally-trained tailored network and perform backpropagation on the converted SNN. The pre-trained word embeddings are also transformed for SNN based text classification. Pre-trained word embeddings are projected into vectors with positive values after proper normalization and shifting, which can be used to initialize tailored networks and converted to spike trains as input of SNNs.

If not considering the contributions of SNN for NLP tasks, the transformation from ANN to SNN is not a new idea actually. This paper is well-written, and the structure is clear for readers to follow.

**Summary Of The Review:**

This paper proposes a new scheme for training the SNN model to achieve text classification in the NLP community. Due to the fact that it is hard to directly train SNN for language tasks using the error backpropagation through time. The authors start with an SNN converted from a normally-trained tailored network and perform backpropagation on the converted SNN. The pre-trained word embeddings are also transformed for SNN based text classification. Pre-trained word embeddings are projected into vectors with positive values after proper normalization and shifting, which can be used to initialize tailored networks and converted to spike trains as input of SNNs. The writing of this paper is good. It is clear for readers to follow and read. but I doubt the significance of the transformation from ANN to SNN model.

---

> ### Author Response · Authors · 2022-11-13
> **Response to Reviewer fFGu.**
>
> Thank you for your valuable comments.
>
> Q1: After a series of operations, the ANN is transformed into SNN version, however, the accuracy and running efficiency are all dropped. A natural question is do we have to conduct such a transformation?
>
> R1: Excessive energy consumption is a major impairment to more wide-spread applications of artificial neural networks (ANNs). Spike-based neuromorphic hardware holds promise to alleviate this problem by more energy-efficient implementations of ANNs than specialized hardware such as GPUs. It has been reported that improvements in energy consumption of up to 2~3 orders of magnitude when compared to conventional ANN acceleration on embedded hardware [1][2][3]. However, deploying high-performing spiking neural networks (SNNs) on such hardware remains a great challenge due to the lack of efficient training algorithms. In this study, we show that combining shallow training (i.e., conversion-based approach) and fine-tuning results in competitive spiking network performance on multiple language benchmarks. SNNs still lag behind ANNs in terms of accuracy, but the gap is narrowing, and can even vanish on some vision tasks. However, there have been very few works that have demonstrated the efficacy of SNNs in NLP tasks. We demonstrate that well-trained spiking neural networks can achieve comparable results to their ANN counterparts on 6 text classification datasets and for both English and Chinese languages.
>
> Q2: The Transformer networks achieve remarkable performance in the NLP tasks, would people select the SNN for their application? I doubt this selection.
>
> R2: Deep neural networks (DNNs) have achieved remarkable results in different applications including language processing tasks, while SNNs are still in the developmental stage and more advanced learning algorithms are expected to train them better. Through intensive research on SNNs in recent years, the performance gap between DNNs and SNNs is constantly decreasing, especially for vision tasks. Our work shows how encoding pre-trained word embeddings as spike trains and training with the two-step recipe (conversion + fine-tuning) can yield competitive performance on multiple text classification benchmarks, thereby giving us a glimpse of how learning algorithms can empower neuromorphic technologies for energy-efficient and ultralow-latency language processing in the future. SNNs cannot currently outperform traditional DNNs on the datasets that were created to train and evaluate conventional DNNs (they use continuous values). Such data should be converted into spike trains before it can be feed into SNNs, and this conversion might cause loss of information and result in a reduction in performance. Therefore, the comparison is indirect and unfair. New datasets that have properties which are compatible with SNNs are expected to be available in the near future, such as those obtained by event-based cameras [4] or the spiking activities that are recorded from biological nervous systems [5], which could be difficult for classical DNNs.
>
> Q3: Not sure about the reproducibility of this work.
>
> R3: We had submitted the source code of the proposed training algorithm with our paper, and plan to release the source code on GitHub upon acceptance. A reproducibility statement was also added in the revised version.
>
> Reference:
>
> [1] Mostafa Rahimi Azghadi, Corey Lammie, Jason K Eshraghian, Melika Payvand, Elisa Donati, Bernabe Linares-Barranco, and Giacomo Indiveri. Hardware implementation of deep network accelerators towards healthcare and biomedical applications. IEEE Transactions on Biomedical Circuits and Systems, 14(6):1138–1159, 2020.
>
> [2] Enea Ceolini, Charlotte Frenkel, Sumit Bam Shrestha, Gemma Taverni, Lyes Khacef, Melika Payvand, and Elisa Donati. Hand-gesture recognition based on EMG and event-based camera sensor fusion: A benchmark in neuromorphic computing. Frontiers in Neuroscience, 14, 2020.
>
> [3] Mike Davies, Andreas Wild, Garrick Orchard, Yulia Sandamirskaya, Gabriel A Fonseca Guerra, Prasad Joshi, Philipp Plank, and Sumedh R Risbud. Advancing neuromorphic computing with Loihi: A survey of results and outlook. Proceedings of the IEEE, 109(5):911–934, 2021.
>
> [4] Ramesh, B., Yang, H., Orchard, G. M., Le Thi, N. A., Zhang, S., & Xiang, C. DART: Distribution aware retinal transform for event-based cameras. IEEE Transactions on Pattern Analysis and Machine Intelligence, 8828, 1, 2019.
>
> [5] Maggi, S., Peyrache, A., & Humphries, M. D. An ensemble code in medial prefrontal cortex links prior events to outcomes during learning. Nature Communications, 9(1), 2018.

---

### Author Response · Authors · 2022-11-13
**The major changes have been made in the revised version**

We thank the reviewers for your insightful comments, which helped us to significantly improve the manuscript. The following major changes have been made in the revised paper:

(1) We have reported the standard deviation in Tables 1, 3, and 4. All the reported average accuracies were obtained over 10 runs with different random initialization for each setting. We found that the standard deviation decreased to 0.33 from 0.65 (almost halved) after the fine-tuning (see Table 1), compared to the simply converted SNNs. The fine-tuning also helps to reduce the number of time steps required to achieve reasonable performance (see Appendix A.4), which shows that the computational cost invested at the fine-tuning phase is worth it.

(2) We have added some text to explain the reason why the clean accuracy of SNNs reported in Table 2 is slightly lower than those in Table1. For the fair comparison, the ensemble method (see Subsection 4.2 for details) was not used in the SNNs when they were evaluated under adversarial attacks since existing studies show that ensemble methods can be used to improve the adversarial robustness. Besides, following the widely-used evaluation setting, we randomly sampled 1,000 examples from each test set to evaluate the models' adversarial robustness because it is prohibitively slow to attack the entire test set.

(3) We have conducted a set of experiments to understand the how the choice of the number of time steps impacts the accuracy of SNNs (see Appendix A.4). We found that the fine-tuned SNNs using 50 time-steps outperform all the converted SNNs without the fine-tuning including those using 80 time-steps, indicating that the proposed fine-tuning method can significantly speed up the inference time and reduce the energy consumption while maintaining the accuracy.

(4) We have compared theoretical energy consumption of SNNs and TextCNNs on 6 different text classification test datasets in Appendix A.3. We show that the SNNs can reduce more than 10 times the energy consumption on average, compared to conventional TextCNNs.

(5) All the reviewers’ comments have been addressed in the revised version.

(6) We have revised the paper thoroughly and carefully. Thank you.

---

### Decision · Program_Chairs · 2023-01-20

**Decision:**

Accept: poster

**Justification For Why Not Higher Score:**

some doubts from the reviewers and AC

**Justification For Why Not Lower Score:**

reviewer and AC concensus

**Metareview: Summary, Strengths And Weaknesses:**

The paper proposes and tested a "conversion + fine-tuning" method for spiking neural networks (SNN). They converted from TextCNN to SNN and demonstrated comparable performance as TextCNN. They also showed more adversarial robustness due to the conversion. Several reviewers objected that the results are no better than TextCNN and that one still needs to train the original TextCNN model. The authors were pretty clear that this was an inference-time procedure, and it is valid to argue that inference-time saving is the most important for a model (perhaps except for research). Since SNN is a new area, the AC and reviewers agree that the paper deserves the benefit of the doubt for proposing an effective conversion procedure.

In addition to some of the doubt raised, the benchmark and model are fairly simple and both the base and converted performances are comparable to bag of words . So the results are vulnerable to the objection that one does not need to do too good a job at the conversion. (The first citation  (Krizhevsky et al., 2017) also appears to have the wrong year. )





**Note From Pc:**

if the above contains the word "oral" or "spotlight" please see: "oral" presentation means -> notable-top-5% and "spotlight" means -> notable-top-25%. As stated in our emails, we are disassociating presentation type from AC recommendations

**Summary Of Ac-Reviewer Meeting:**

insufficient responses. Chatted with individual reviewers about particular points.